# Red-Emitting Carbon Quantum Dots for Biomedical Applications: Synthesis and Purification Issues of the Hydrothermal Approach

**DOI:** 10.3390/nano13101635

**Published:** 2023-05-13

**Authors:** Barbara La Ferla, Barbara Vercelli

**Affiliations:** 1Dipartimento di Biotecnologie e di Bioscienze, Università degli Studi di Milano-Bicocca, Piazza della Scienza, 2, 20126 Milano, Italy; barbara.laferla@unimib.it; 2Istituto di Chimica della Materia Condensata e di Tecnologie per l’Energia, CNR-ICMATE, Via Cozzi, 53, 20125 Milano, Italy

**Keywords:** red-emitting carbon quantum dots, hydrothermal approaches, sustainable strategies

## Abstract

The possibility of performing the synthesis of red-emitting carbon quantum dots (r-CDs), in a well-controllable, large scale and environmentally sustainable way is undoubtedly of fundamental importance, as it will pave the way to their employment in advanced medical large-scale applications. Knowledge of the difficulties involved in producing r-CDs with reproducible optical, structural, and chemical characteristics, might help in their large-scale production, making the process standardizable. In this work, we present an experimental study, also supported by results reported in the literature, on the issues encountered during the synthesis and post-synthesis purification treatments of r-CDS. We focused on the hydrothermal approach as it was found to be more suitable for future large-scale industrial applications. We propose three synthetic strategies and observed that employing p-phenylenediamine (p-PDA), as a precursor, the synthetic process showed low efficiency with low yields of r-CDs, large amounts of unreacted precursor, and reaction intermediates. Changing reaction parameters does not improve performance. The r-CDs obtained using citric acid (CA) and urea, as precursors, resulted to be sensitive to pH and difficult to separate from the reaction mixture. Furthermore, the proposed synthetic strategies show that the hydrothermal preparation of r-CDS requires approaches that are not fully sustainable.

## 1. Introduction

Carbon quantum dots (CDs) are known as zero-dimensional carbonaceous nanomaterials with incomparable optical properties [1,2]. Since their accidental discovery in 2004, they have become a central topic of the scientific community because of their peculiar properties, which make them suitable materials for application in numerous fields, including in vitro and in vivo bioimaging [3,4,5], the development of antibacterial and antiviral agents [6,7], sensing [8], and optoelectronic devices [9,10]. In many cases, CDs are considered as graphitic carbon cores embedded in amorphous shells with different surface functionalities, such as carboxyl, amide, hydroxyl, and carbonyl moieties, which allow their derivatization with various entities (i.e., enzymes, etc.) [11]. CDs exhibit an excellent intrinsic fluorescence, which can be tuned from the visible to NIR, and are also characterized by extremely low toxicity and excellent biocompatibility crucial for real-world biological applications.

Red-emitting CDs (r-CDs) result particularly suitable for biomedical applications because red light shows low absorption, deep tissue penetration, and weak background biological fluorescence [12,13,14]. The preparation of CDs in the far-red to NIR region has been considered the basic and fundamental requirement for practical application in the biological and medical fields [15,16]. Therefore, the development of efficient and reproducible synthetic routes for r-CDs production can advance their applications in many scientific and industrial areas. The literature is full of research papers dealing with synthetic approaches of r-CDs for their use in the biomedical field. The most recent ones are the studies of Luo et al. [17] and Li et al. [18] reporting, respectively, the hydrothermal synthesis of r-CDs for application in the detection of hematin in red cells and in the imaging of lysosomes. On the other hand, Jia et al. [19] report the synthesis of water-soluble r-CDs employed in the one proton and two protons imaging of various samples both in vitro and in vivo. Particularly interesting are the works of Dong et al. [20] and Oluwafemi et al. [21] which report cost-effective green and sustainable approaches for the preparation of r-CDs. Lin et al. [12], in their recent review work on the tuning of the optical properties of r-CDs and their applications, observed that the development of simple methods to manufacture r-CDs with uniform size, structure, composition, and surface state could be an efficient way to solve the problems related to the low fluorescence intensity, especially shown in the long wavelength range. It seems that the synthetic strategies reported for far-red to NIR CDs lack a rational design, and most of those reported are semi-empirical. The synthesis of r-CDs seems to be a sort of black box, and it is still difficult to obtain CDs with specific optical properties by rationally choosing the precursors and reaction conditions, and very few reproducible synthetic strategies have been published. Two recent review papers [12,22] on synthetic strategies for r-CDs, try to provide some sort of rationalization by focusing on key factors such as solvent, precursors, and purification methods. In particular, Bagheri et al. [22] highlight the application of statistical and computational methods for the optimization of key parameters and the prediction of optical features.

Starting from these considerations, in the present work we report an experimental study, also supported by literature data, which describes the problems encountered during the hydrothermal synthesis and post-synthesis purification treatments of r-CDs. We chose the hydrothermal method because its results more suitable for future large-scale industrial applications. We propose three synthetic strategies depicted in Figure 1 and we observe how the reaction parameters (solvent, co-precursor, and temperature) can influence the reaction efficiency and the optical characteristics of the obtained CDs. We also provide information about the applicability of the purification methods in terms of reaction efficiency and sample damage. Finally, we discuss the issues of obtaining r-CDs with sustainable approaches.

## 2. Materials and Methods

All used chemicals and solvents were reagent grade and used as received. pH buffers (3, 4, 7, 9, 12) for UV-vis analyses were purchased from Carlo Erba (Milan, Italy). Thin-layer-chromatography (TLC) was performed on silica gel 60F_254_ plates (Merck, Darmstadt, Germany) and visualized with UV detection (254 nm and 365 nm).


**r-CDs Synthesis**


*p-phenylenediamine*—r-CDs were prepared according to a previously reported synthesis route [23]. A total of 0.25 g (2.3 mmol) of p-phenylene diamine were dissolved in 25 mL of selected solvent (EtOH or MeOH or deionized water), transferred into a Teflon-lined stainless-steel reactor (100 mL) and heated at 180 °C for 24 h. The reactor was then allowed to cool down to room temperature, and the solution was filtrated through a microporous membrane (0.22 μm) to remove the large particles. The obtained filtrate was concentrated and purified by silica gel column chromatography using a dichloromethane/EtOH (9.5:0.5) mixture as eluent.

*p-phenylenediamine and CA*—CDs were prepared according to the reported synthesis route [24]. A total of 1.8 g (17 mmol) of p-phenylenediamine and 3.2 g (17 mmol) of citric acid (CA) were dissolved in 50 mL of deionized water and stirred for 15 min; then the solution was transferred into a Teflon-lined stainless-steel reactor (100 mL) and heated at 210 °C for 10 h. The reactor was allowed to cool down to room temperature, and the solution was filtrated through a microporous membrane (0.22 μm).

*p-phenylenediamine and urea*—CDs were prepared according to the reported synthesis route [25]. A total of 0.2 g (1.85 mmol) of p-phenylenediamine and 0.2 g (3.3 mmol) of urea were dissolved in 50 mL of deionized water and stirred for 15 min; then the solution was transferred into a Teflon-lined stainless-steel reactor (100 mL) and heated at 160 °C or 210 °C for 10h. The reactor was allowed to cool down to room temperature, and the solution was filtrated through a microporous membrane (0.22 μm).

*CA and urea*—CDs were prepared according to the reported synthetic route [26]. 1.3 g (7 mmol) of CA and 1.3 g (21 mmol) of urea were dissolved in 20 mL of solvent (formamide or EtOH or deionized water) and stirred for 15 min; then the solution was transferred into a Teflon-lined stainless-steel reactor (100 mL) and heated at 180 °C for 12 h. The reactor was then allowed to cool down to room temperature, and the solution was filtrated through a microporous membrane (0.22 μm). The CDs mixtures obtained with EtOH and water were precipitated with CH_3_CN and EtOH, respectively, followed by centrifugation. The re-dispersion/precipitation procedure was repeated three times to ensure the effective removal of precursor residues and small derivatives. The CDs mixture obtained with formamide as solvent was purified following a literature-recommended procedure [4] employing an ion-exchange column (dowex 1 × 8 chloride form). A plastic column was filled with 8 mL of dowex 1 × 8 chloride form (100–200 mesh), equilibrated with 0.5 m hydrochloric acid, and copiously washed with deionized water to a neutral ph. then 200 μL of the CDs mixture was carefully loaded on the column. CDs fractions were respectively separated employing deionized water, and HCl solution 1 M, 5 M and 10 M.


**Characterization Techniques**


UV-vis spectra were collected with a Perkin Elmer Lambda 35 spectrometer; FTIR spectra were measured by infrared microscope Varian 610-IR coupled to the Varian 670-IR FTIR spectrometer (Varian, Australia Pty Ltd., Melbourne, Australia), equipped with a mercury cadmium telluride nitrogen cooled detector.

The continuous wave PL spectra were recorded at 25 °C using a Cary Eclipse Fluorescence Spectrometer (Agilent, Santa Clara, USA).

^1^HNMR spectra were recorded using a Bruker 400 MHz at 25 °C. Chemical Shifts (δ) are reported in ppm downfield from the residual solvent peak. NMR data processing was performed with MestReNova v14.2.0 software (Mestrelab Research, Santiago de Compostela, Spain).

## 3. Results and Discussion

In the first proposed synthetic strategy, we repeated a literature-reported hydrothermal route [24] based on an ethanol solution of p-PDA heated at 180 °C in an autoclave for 24 h (Figure 1A). Surprisingly and in contrast with reported data, the reaction mixture contained along with the red-emitting component, other emitting ones (see TLC in Appendix A). After purification through silica gel column chromatography (eluent DCM/EtOH 9.5:0.5), four fractions were separated, which were analyzed with spectroscopic techniques (UV-vis, FT-IR and ^1^H-NMR). According to TLC, the UV-vis spectra (Figure 1a) show that the first fraction eluted (fraction A) mainly consists of un-reacted p-PDA.

The main absorption bands of p-PDA at 244 nm, and at 310 nm are present, and can be, respectively, ascribed to the π–π* transition of the C=C bonds of the ring, and the n–π* transition of the C=N bonds. A third band is also present at 285 nm (1 in Figure 1a), ascribable to the π–π* transition of condensed rings [27] and may indicate the presence of graphitic structures [28,29]; the presence of aromatic structures other than p-PDA is also confirmed in the NMR spectra by the presence of proton signals between 6.5 and 8.5 ppm (Appendix A). The UV-vis spectra of all the remaining fractions (Figure 1a) do not present the p-PDA absorption bands. The spectrum of the second fraction (fraction B) presents band 1, ascribable to the presence of the graphic cores [28,29], and a second band at 507 nm (2 in Figure 1a), which, as will be treated below, is responsible for the red emission of the sample. A shoulder at 256 nm 3 in Figure 1a) is also present, which may be ascribed to the π–π* transition of fused rings, and may indicate the presence of both short polynuclear aromatic systems, bonded to the graphitic cores and small graphitic structures in the growth stage (i.e., graphene sheets or graphene CDs [28,29]). In agreement with literature data [23] fraction B corresponds to the r-CDs, and it results to be the less abundant component of the reaction mixture: only 5% yield. The spectra of the third and fourth fractions (fractions C and D) are similar. Compared to fraction B, in addition to band 1, shoulder 3 turns into a band, band 2 shifts of ca 20 nm to lower wavelengths and a new band at 415 nm appears. Photoluminescence emission (PL-emission) determinations show that fractions B, C, and D excited at 510 nm (Figure 1b) present a band at ca 600 nm (red-emission), whose intensity progressively decreases going from fraction B to fraction D. The stokes shifts are 90 nm for fraction B, and 100 nm for fraction C and D. The estimated PL QY (see Appendix A for more information) of fraction B is ca 27%, in agreement with literature reported data [23]. Furthermore, fraction B proved to be stable both in solid and in EtOH solution for over 1 month. FT-IR determinations (Figure 1c) show that at the edges of the basal planes of the graphitic cores of fractions B, C and D are present short polynuclear aromatic systems and aromatic amines. In particular, the short polynuclear aromatic systems bands (red-starred in Figure 1c) are the aromatic C-H stretching at 3030 cm^−1^, the skeletal vibrations involving the C to C stretching within the ring at 1512 cm^−1^ and 1441 cm^−1^, the out-of-plane C-H bending at 830 cm^−1^ [27]; the amine bands (blue-starred in Figure 1c) are the asymmetric N-H stretching at 3420 cm^−1^, the symmetric N-H stretching at 3329cm^−1^ and 3204 cm^−1^ and the N-H bending at 1633 cm^−1^ [27]. The ^1^H NMR spectra of fractions B, C, and D also confirm the presence of the signals between 6.7 and 8.5 ppm that can belong to polycyclic aromatic systems containing conjugated pyrroles and pyridines (Appendix A). FT-IR spectra of all three fractions also show the C-H stretching bands of aliphatic systems at 2910 cm^−1^ and 2850 cm^−1^ (black starred in Figure 1c), which may be ascribable to a possible role played by the solvent EtOH in the CDs formation, the presence of non-aromatic protons is also confirmed by NMR signals below 4 ppm. In particular, the co-presence of the FT-IR bands typical of esters, such as the C=O stretching band at 1740 cm^−1^ and the C-O stretching band at 1250 cm^−1^ (green starred in Figure 1c) [27], may suggest that the solvent causes the formation of esters functional groups, bonded to the graphitic cores. The presence of ethyl esters is evident in NMR spectra of fractions B and C testified by the quartet at 3.6 ppm (-OCH_2_CH_3_) and 1.2 ppm (-OCH_2_CH_3_).

Given the possible role played by the reaction medium in the formation of r-CDs, we performed reaction route A (Figure 1) using MeOH and water, respectively, and keeping unchanged all other reaction parameters. The aim was to obtain the r-CDs as the only component in large amounts and make the process more sustainable, using a greener solvent. Unfortunately, the UV-vis spectra of the reaction mixtures (Appendix A) show that the reaction performance seems to decrease going from EtOH to water. The absorption bands 1 (starred in Appendix A and ascribed to the presence of graphitic cores [27,28,29]) and 2 (responsible for the red emission) progressively decrease in intensity leaving only the typical absorption bands of un-reacted p-PDA. Furthermore, TLC (Appendix A) of all reaction mixtures shows, that there are no considerable differences between the products obtained from the syntheses with EtOH and MeOH, and the product obtained from the reaction with water does not present the red-emitting component, but only the un-reacted p-PDA.

In the light of the above-observed difficulties in obtaining r-CDs using p-PDA in water and trying to overcome them, we performed two more syntheses, where we employed two bio-synthesizable and low-cost co-precursors, such as CA and urea, respectively (Figure 1B,B′), together with p-PDA and water. The UV-vis spectra of the reaction mixtures obtained from both synthetic routes (Appendix A) show the typical π–π* and n-π* transition bands, at 240 nm and 300 nm, respectively, ascribed to un-reacted p-PDA, which cover the possible bands of CDs. In particular, the spectrum of the reaction mixture obtained using CA does not present absorption bands in the low energy absorption region and the solution under a 365 nm UV lamp shows green emission (Appendix A), with no red components. On the other hand, the reaction mixture obtained using urea presents a further weak absorption band at ca 495 nm (Appendix A), which may indicate the presence of a red component. However, its TLC (Appendix A) shows the presence of at least four more fractions together with the red one and the un-reacted p-PDA. Increasing the reaction temperature from 160 °C to 210 °C does not improve the reaction performance, in terms of complete conversion of p-PDA, and/or reduction of the number of mixture components in favor of the red one.

Therefore, we stopped employing p-PDA as the precursor, and we performed three new syntheses (Figure 1C) employing CA and urea, as precursors, with different reaction solvents: water, EtOH, and formamide. The obtained reaction mixtures (Figure 2A,B) clearly show the influence of the solvent on the preparation of CDs. Only the samples obtained with formamide are red-colored (Figure 2A) and the red emission under UV-lamp is difficult to observe, because the samples showed to be a mixture of different emitting components (Figure 2B).

On the other hand, the mixtures obtained from water and EtOH are pale yellow and golden yellow colored, respectively, showing blue and pale green emissions, under a UV lamp. The UV-vis spectrum (Figure 3a) of the formamide sample shows an absorption band at 263 nm, which may indicate the presence of graphitic cores [27,28,29], and a second absorption peak at 554 nm with a shoulder at 520 nm, responsible of the red-emission.

PL-determinations show that the PL-emission bands peaked at ca 600 nm (red emission, Figure 3b), upon 520 nm and 560 nm excitation with Stokes shifts of ca 80 nm and 40 nm, respectively. The UV-vis spectra of the samples obtained from water and EtOH (Figure 3a) are similar, showing an n–π* transition band at 330 nm and 340 nm, respectively, due to C=O and C=N bonds. They also proved to be easily purified through precipitation/centrifugation cycles, giving solids that could be redispersed in water and EtOH, respectively [30,31]. On the other hand, the separation of the red component from the formamide mixture was revealed to be not so easy. We followed a literature-recommended procedure [26], based on an ion-exchange column employing HCl at increasing concentrations, as eluent, and we discovered that the red component deteriorates during the purification process, because it showed to be sensitive to solution pH. In particular, as Figure 3c shows, the intensity of the low energy absorption band decreases in the low pH region, resulting in the loss of the red PL-emission. The observed phenomenon, in contrast with what was observed for the CDs obtained from water [30], was found to be irreversible. The resulting red component was stable in neutral and alkaline media (pH 7 and 9) and deteriorates in acidic media (pH ≤ 4).

The above-reported results enable some considerations on the application of hydrothermal synthetic approaches in the production of red-emitting CDs. Starting from the first proposed strategy (Figure 1A), we can state that the reaction reported in the literature [22] did not go to completion with the conversion of p-PDA into r-CDs (fraction B). UV-vis analyses (Figure 1a) show the co-presence of un-reacted p-PDA (fraction A) together with two more species (fractions C and D), which may be considered two growth-stage intermediates of condensation/graphitization of p-PDA. It is noteworthy that the UV-vis band 3 at 256 nm, in fraction B is a shoulder, while in fractions C and D is a defined peak. On the other hand, band 2 splits into two peaks, and shifts to higher energies. We above-ascribed band 3 to fused rings of short polynuclear aromatic systems [27], whose presence in all fractions is also confirmed by FT-IR and ^1^HNMR determinations. Thus, we may suppose that these short polynuclear aromatic systems are the first stage species of the condensation/cyclization processes of conversion of p-PDA into more complex graphitic structures. The observed decrease of the intensity of band 3, becoming a shoulder in fraction B and leaving only band 1 (ascribed to the presence of graphitic cores [27,28,29], Figure 1a), may indicate the progressive growth of these short polynuclear aromatic systems and their conversion into graphitic structures. Furthermore, the proposed synthetic approach shows that fraction B, i.e., r-CDs, resulted in being the less abundant component (estimated yield of 5%) indicating that the synthetic strategy is not efficient, and not suitable for industrial applications. We observed that neither increasing reaction temperature and time, respectively, nor working under an inert atmosphere (Ar) improves the reaction performance. Furthermore, we observed that the solvent is also involved in CDs formation. In the case of EtOH, FT-IR and NMR determinations suggest that the solvent may be responsible for the formation of ester groups bonded to the graphitic cores. Changing EtOH with MeOH and water does not improve the reaction performance in favor of the red component; besides in water, p-PDA seems to remain un-reacted.

The second proposed strategy (Figure 1B,B′) shows that the addition of co-precursors, such as CA and urea, does not improve the p-PDA hydrothermal conversion into r-CDs. In particular, in both cases we observed the dominant presence of unreacted p-PDA, and, in the case of CA, the obtained product shows green emission under UV lamp.

In the third proposed synthetic strategy (Figure 1C), it is noteworthy the influence of the solvent on the preparation of CDs. The red component is present only in the synthetic route employing formamide, and it is absent in the ones employing water or EtOH. Furthermore, the red component is revealed to be sensitive to solution pH and deteriorates in acidic media. Thus, the separation with a literature-recommended procedure [26] based on an ion exchange column employing HCl, caused its damage.

Finally, taking into account the three proposed hydrothermal synthetic routes it results that the preparation of r-CDs is not an environmentally friendly, facile, and low cost-cost process. In agreement with what was observed by Bagheri et al. [22], considering the first two proposed approaches, r-CDs are obtained by employing a toxic, high-cost precursor, such as p-PDA. On the other hand, in the third approach, where p-PDA was replaced by two bio-synthesizable and low-cost precursors (CA and urea), to obtain r-CDs the use of formamide as solvent is necessary, which is not properly green and sustainable. Furthermore, the post-synthesis purification treatment was complex with small reaction yields and/or damage of the red component. In the present case, the dialysis route was not taken into account as a purification strategy, because the products of all the studied synthetic strategies resulted in mixtures of different emitting components (see also the TLC reported in the supporting information). The employment of chromatographic techniques was found to be more suitable for their separation/purification.

## 4. Conclusions

In conclusion, in the present work, we reported the limitations and problems encountered in employing hydrothermal synthetic strategies for the sustainable hydrothermal preparation of r-CDs. In general, we observed that using p-PDA, as starting material, the reaction process showed low efficiency with low yields of r-CDs, a large amount of un-reacted p-PDA, and reaction intermediates. Changing the reaction parameters (solvent, temperature, co-precursors) does not improve the reaction performance. On the other hand, the observed pH dependence of r-CDs complicates the post-treatment separation steps, causing damage to the red component.

The results above-described, in agreement with what was reported by Bagheri et al. [22], show that the preparation of r-CDs in a well-controllable, large scale and sustainable way presents some problems, the knowledge of which could be useful to improve their synthetic strategies, allowing the scale-up for industrial applications. For example, knowledge of the low efficiency of the synthesis process can help both to improve the process itself and to find sustainable strategies to recycle and reuse the unreacted precursors and reaction by.productus. Undoubtedly, the possibility to perform the synthesis of long-wavelength CDs, controlled according to the purpose, is of paramount importance because it will pave the way to their employment in a wide range of advanced medical applications. To this end, further exploration is still ongoing.

## Data Availability

Data is contained within the article and Appendix A.

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
