# Peer review of "Red-Emitting Carbon Quantum Dots for Biomedical Applications: Synthesis and Purification Issues of the Hydrothermal Approach"

_nanomaterials, 2023, doi:10.3390/nano13101635_

Round 1

Reviewer 1 Report

The manuscript describes the synthesis of red emissive carbon nanodots. However, I found it much lower than the basic requirement for publication as a scientific paper. Major concerns are:

1. The main body of the manuscript is not consistent with its title. 

2. Too limited data to illustrate a whole story of red emitting carbon nanodots.

3. The photo shown in Figure 2 is not consistent with the PL spectra shown in Figure 1.

4. Figure 3 is missing.

English quality is not the most important issue for this manuscript.

Reviewer 2 Report

The manuscript under consideration is a very important piece of "negative result", yet rarely reported in the literature. Moreover, the issue of the CDs yield it not so often touched in the research reports. It has been a great surprise for me also when I directly measured the yield of CDs in a similar synthesis, reported in many publications as facile and straightforward, and found that it was hardly of 1%. So, in general I strongly support publishing of this report. Although it is probably not so comprehensive as it could be, it is about to start an important discussion likely to be further extended by this team and other researchers.

Being in general positive about accepting this manuscript, I would like to list below some suggestions to improve it.

1. Please carefully reconsider English spelling and grammar throughout the entire text. The mistakes unfortunately include the instrument manufacturer (Agilet instead of Agilent, line 134), the chemicals ('phenilene' instead of phenylene in lines 100 and 106), and scientific terms (mash instead of mesh, line 122). There are also numerous less important mistakes, but at least professional terminology should be double-checked.

2. Please reconsider Scheme 1. The lists of alternative conditions (different solvents, temperatures) look like sequential treatment under different conditions.

3. Considering the syntheses (p. 3): I have noticed that the authors used the same reactor (100 mL) although the volume of the reaction mixture was different depending on the synthesis (~21 to ~ 53 mL). In my view, the filling degree of the reactor can affect the synthesis by changing the pressure in the reactor. Do the reactors filling used in this study match the conditions reported in the source references?

4. The authors have not mentioned dialysis as a route to CDs purification. In my experience, it is a powerful yet time-taking procedure to isolate CDs free of low-molecular admixtures. It would be nice if the authors add some discussion on this point (I do not mean repeating the experiment, just discussion based on reported studies would add to the manuscript content).

5. Regarding the discussion of the synthesis from PDA: please explain how the yield was calculated (line 186) and report the excitation wavelength (lines 189-194).

6. Please check the wavelength of the UV lamp (lines 235 and 257). 365 nm is more common.

7. Sorry, I cannot see any "sort of red emission" in Fig 2B as mentioned in line 247. Moreover, since the sample is red itself, this "red emission" could be just illumination of the sample with a broad blue-green emission band. Please reconsider this description and the figure.

8. I cannot see Fig. 3 at all - just a blank space.

9. Since Fig. 3 is not seen, it is hard to follow its discussion, but there is an immediate question regarding the phenomenon described in lines 298-303: are the pH-induced changes reversible?

Please see i. 1 in the general review.

Author Response

see tha attached file

Reviewer 3 Report

The article “Red-emitting Carbon Quantum Dots for Biomedical Applications: Synthesis and Purification Issues” describes the attempts of authors to obtain red-emitting carbon quantum dots (r-CDs). The authors used straight forward synthesis route in Teflon lined-up autoclave for characterization using PL, UV-Vis and HNMR, and TLC. The article focuses on PL as the title suggests but what is missing is the sizes of the nanoparticles obtained. Can authors perform TEM measurements or at least suggests the sizes according to light color?  The second problem is missing Figure 3 I cannot relate to something which I cannot see. Please correct everything and send the article again.

Corrections:

Fig 2 label B should be lower to not cover the image.

Fig 3 is missing in the pdf file. Please append the entire file once again

Line 48 semi -> hemi-empirical [12]

Round 2

Reviewer 1 Report

1.     The title notified that these r-CDs are for biomedical applications, while no related investigation have been casted.

2.     This manuscript might focus on the synthesis and purification. But the authors still do not delineate how this work is novel, or what is the difference, in comparison with previous papers.

3.     To provide a broad scenario of its current research status, some important advances in the emissive CDs closely related to the topic of this manuscript, are strongly suggested to be included in the introduction section. e.g., ACS Sustainable Chem. Eng. 2023, 11(17), 6535–6544; Advanced Science 2021, 8(6), 2003433; Nano Research 2021, 14(7), 2231-2240; Nano Research 2020, 13(3), 875-881.

4.     The authors are encouraged to include more specific characterization results of their own CDs products. For example, Raman spectra are suggested to verify the CDs’ structure as complementary of FTIR. And XPS to identify the surface chemical composition which is sciential for biological applications.

5.     Figure 1 and 3 are missing. I cannot evaluate whether the contents are sound and convincing.

Grammar errors and typos need to be double-checked throughout the entire manuscript. For example, “whicht” for “which”, etc.

Author Response

see the attache file

Reviewer 3 Report

The authors corrected the article according to previous suggestions. The answers provide are sufficient. I propose the article for publication.

Author Response

We thank a lot the reviewer for the suggestions